# Unsupervised Co-Learning on $\mathcal{G}$-Manifolds Across Irreducible Representations

**Yifeng Fan**[1] **Tingran Gao**[2] **Zhizhen Zhao**[1]
[1]University of Illinois at Urbana-Champaign  [2]University of Chicago
{yifengf2, zhizhenz}@illinois.edu  tingrangao@galton.uchicago.edu

## Abstract

We introduce a novel co-learning paradigm for manifolds naturally admitting an action of a transformation group $\mathcal{G}$, motivated by recent developments on learning a manifold from attached fibre bundle structures. We utilize a representation theoretic mechanism that canonically associates multiple independent vector bundles over a common base manifold, which provides multiple views for the geometry of the underlying manifold. The consistency across these fibre bundles provide a common base for performing unsupervised manifold co-learning through the redundancy created artificially across irreducible representations of the transformation group. We demonstrate the efficacy of our proposed algorithmic paradigm through drastically improved robust nearest neighbor identification in cryo-electron microscopy image analysis and the clustering accuracy in community detection.

## 1 Introduction

Fighting with the *curse of dimensionality* by leveraging low-dimensional intrinsic structures has become an important guiding principle in modern data science. Apart from classical structural assumptions commonly employed in sparsity or low-rank models in high dimensional statistics [63, 11, 12, 49, 2, 64, 67], recently it has become of interest to leverage more intricate properties of the underlying geometric model, motivated by algebraic or differential geometry techniques, for efficient learning and inference from massive complex datasets [15, 16, 44, 46, 8]. The assumption that high dimensional datasets lie approximately on a low-dimensional manifold, known as the *manifold hypothesis*, has been the cornerstone for the development of manifold learning [62, 52, 18, 3, 4, 5, 17, 57, 66] in the past few decades.

In many real applications, the low-dimensional manifold underlying the dataset of high ambient dimensionality admits additional structures that can be fully leveraged to gain deeper insights into the geometry of the data. One class of such examples arises in scientific fields such as *cryo-electron microscopy* (cryo-EM), where large numbers of random projections for a three-dimensional molecule generate massive collections of images that can be determined only up to in-plane rotations [59, 72]. Another source of examples is the application in computer vision and robotics, where a major challenge is to recognize and compare three-dimensional spatial configurations up to the action of Euclidean or conformal groups [28, 10]. In these examples, the dataset of interest consists of images or shapes of potentially high spatial resolution, and admits a natural group action $g \in \mathcal{G}$ that plays the role of a nuisance or latent variable that needs to be "quotient out" before useful information is revealed.

In geometric terms, on top of a differentiable manifold $\mathcal{M}$ underlying the dataset of interest, as assumed in the manifold hypothesis, we also assume the manifold admits a smooth *right action* of a Lie group $\mathcal{G}$, in the sense that there is a smooth map $\phi : \mathcal{G} \times \mathcal{M} \to \mathcal{M}$ satisfying $\phi(e, m) = m$ and $\phi(g_2, \phi(g_1, m)) = \phi(g_1 g_2, m)$ for all $m \in \mathcal{M}$ and $g_1, g_2 \in \mathcal{G}$, where $e$ is the identity element of $\mathcal{G}$. A *left action* can be defined similarly. Such a group action reflects abundant information about the symmetry of the underlying manifold, with which one can study geometric and topological

properties of the underlying manifold through the lens of the orbit, stabilized, or induced finite- or infinite-dimensional representations of $\mathcal{G}$. In modern differential and symplectic geometry literature, a smooth manifold $\mathcal{M}$ admitting the action of a Lie group $\mathcal{G}$ is often referred to as a $\mathcal{G}$-**manifold** (see e.g. [40, §6], [50, 1, 33] and references therein), and this transformation-centered methodology has been proven fruitful [42, 53, 40, 30] by several generations of geometers and topologists.

Recent development of manifold learning has started to digest and incorporate the additional information encoded in the $\mathcal{G}$-actions on the low-dimensional manifold underlying the high-dimensional data. In [36], the authors constructed a *steerable graph Laplacian* on the manifold of images — modeled as a rotationally invariant manifold (or U (1)-manifold in geometric terms) — that serves the role of graph Laplacian in manifold learning but with naturally built-in rotational invariance by construction. In [38], the authors proposed a principal bundle model for image denoising, which achieved state-of-the-art performance by combining patch-based image analysis with rotationally invariant distances in microscopy [47]. A major contribution of this paper is to provide deeper insights into the success of these group-transformation-based manifold learning techniques from the perspective of *multi-view learning* [56, 60, 37] or *co-training* [7], and propose a family of new methods that systematically utilize these additional information in a systematic way, by exploiting the inherent consistency across representation theoretic patterns. Motivated by the recent line of research bridging manifold learning with principal and associated fibre bundles [57, 58, 22, 20, 19], we point out that to a $\mathcal{G}$-manifold admitting a principal bundle structure is naturally associated as many vector bundles as the number of distinct irreducible representations of the transformation group $\mathcal{G}$, and each of these vector bundles provide a separate "view" towards unveiling the geometry of the common *base manifold* on which all the fibre bundles reside.

Specifically, the main contributions of this paper are summarized as follows: (1) We propose a new unsupervised co-learning paradigm on $\mathcal{G}$-manifold and propose an optimal alignment affinity measure for high-dimensional data points that lie on or close to a lower dimensional $\mathcal{G}$-manifold, using both the local cycle consistency of group transformations on the manifold (graph) and the algebraic consistency of the unitary irreducible representations of the transformations; (2) We introduce the invariant moments affinity in order to bypass the computationally intensive pairwise optimal alignment search and efficiently learn the underlying local neighborhood structure; and (3) We empirically demonstrate that our new framework is extremely robust to noise and apply it to improve cryo-EM image analysis and the clustering accuracy in community detection. Code is available on `https://github.com/frankfyf/G-manifold-learning`.

## 2 Related Work

**Manifold Learning:** After the ground-breaking works of [62, 52], [5, 56, 41] provided reproducing kernel Hilbert space frameworks for scalar and vector valued kernel and interpreted the manifold assumption as a specific type of regularization; [3, 4, 14] used the estimated eigenfunctions of the Laplace–Beltrami operator to parametrize the underlying manifold; [24, 25, 59] investigated into the representation theoretic pattern of an integral operator acting on certain complex line bundles over the unit two-sphere naturally arising from cryo-EM image analysis; [57, 58, 22] demonstrated the benefit of using differential operators defined on fibre bundles over the manifold, instead of the Laplace–Beltrami operator on the manifold itself, in manifold learning tasks. Recently, [20, 19, 23] proposed to utilize the consistency across multiple irreducible representations of a compact Lie group to improve spectral decomposition based algorithms.

**Co-training and Multi-view Learning:** In their seminal work [7], Blum and Mitchell demonstrated both in theory and in practice that distinct "views" of a dataset can be combined together to improve the performance of learning tasks, through their complementary yet consistent prediction for unlabelled data. Similar ideas exploiting the consistency of the information contained in different sets of features has long existed in statistical literature such as canonical correlation analysis [29]. Since then, multi-view learning has remained a powerful idea percolating through aspects of machine learning ranging from supervised and semi-supervised learning to active learning and transfer learning [21, 43, 61, 13, 55, 56, 34, 35]. See surveys [60, 69, 70, 37] for more detailed accounts.

## 3 Geometric Motivation

We first provide a brief overview of the key concepts used in this paper from elementary group representation theory. Interested readers are referred to [54, 9] for more details.

**Groups and Representation:** A *group* $\mathcal{G}$ is a set with an operation $\mathcal{G} \times \mathcal{G} \to \mathcal{G}$ obeying the following axioms: (1) $\forall g_1, g_2 \in \mathcal{G}, g_1 g_2 \in \mathcal{G}$; (2) $\forall g_1, g_2, g_3 \in \mathcal{G}, g_1(g_2 g_3) = (g_1 g_2) g_3$; (3) There is a unique $e \in \mathcal{G}$ called the *identity* of $\mathcal{G}$, such that $eg = ge = g, \forall g \in \mathcal{G}$; (4) $\forall g \in \mathcal{G}$, there is a corresponding element $g^{-1} \in \mathcal{G}$ called the *inverse* of $g$, such that $g g^{-1} = g^{-1} g = e$. A $d_\rho \times d_\rho$-dimensional *representation* of a group $\mathcal{G}$ over a field $\mathbb{F}$ is a matrix valued function $\rho : \mathcal{G} \to \mathbb{F}^{d_\rho \times d_\rho}$ such that $\rho(g_1)\rho(g_2) = \rho(g_1 g_2), \forall g_1, g_2 \in \mathcal{G}$. In this paper, we assume $\mathbb{F} = \mathbb{C}$. A representation $\rho$ is said to be *unitary* if $\rho(g^{-1}) = \rho(g)^*$ for any $g \in \mathcal{G}$ and $\rho$ is said to be *reducible* if it can be decomposed into a direct sum of lower-dimensional representations as $\rho(g) = Q^{-1}(\rho_1(g) \bigoplus \rho_2(g))Q$ for some invertible matrix $Q \in \mathbb{C}^{d_\rho \times d_\rho}$, otherwise $\rho$ is *irreducible*, the symbol $\bigoplus$ denotes the direct sum. For a compact group, there exists a complete set of *inequivalent irreducible representations (**in brevity: irreps**)* and any representation can be reduced into a direct sum of irreps.

**Fourier Transform:** In many applications of interest, the Lie group is compact and thus always admits irreps, and the concept of irreps allows generalizing the Fourier transform to any compact group. By the renowned Peter–Weyl theorem, any square integrable function $f \in L_2(\mathcal{G})$ can be decomposed as

$$f(g) = \sum_{k=0}^{\infty} d_k \mathrm{Tr}\left[ F_k \rho_k(g) \right], \quad \text{and} \quad F_k = \int_\mathcal{G} f(g) \rho_k^*(g) d\mu_g, \tag{1}$$

where each $\rho_k : \mathcal{G} \to \mathbb{C}^{d_k \times d_k}$ is a *unitary irrep* of $\mathcal{G}$ with dimension $d_k \in \mathbb{N}$. This is the compact Lie group analogy of the standard Fourier series over the unit circle. The *"generalized Fourier coefficient"* $F_k$ in (1) is defined by the integral taken with respect to the Haar measure on $\mathcal{G}$.

**Motivation:** Motivated by [38, 36], we consider the principal bundle structures on a $\mathcal{G}$-manifold $\mathcal{M}$. Below we state the definitions of fibre bundle and principal bundle for convenience; see [6] for more details. Briefly speaking, a fibre bundle is a manifold which is locally diffeomorphic to a product space, and a principal fibre bundle is a fibre bundle with a natural group action on its "fibres."

**Definition 1 (Fibre Bundle)** *Let $\mathcal{M}, \mathcal{B}, \mathcal{F}$ be three differentiable manifolds, and let $\pi : \mathcal{M} \to \mathcal{B}$ denote a smooth surjective map between $\mathcal{M}$ and $\mathcal{B}$. We say that $\mathcal{M} \xrightarrow{\pi} \mathcal{B}$ (or just $\mathcal{M}$ for short) is a **fibre bundle** with typical fibre $\mathcal{F}$ over $\mathcal{B}$ if $\mathcal{B}$ admits an open cover $\mathscr{U}$ such that $\pi^{-1}(U)$ is diffeomorphic to product space $U \times \mathcal{F}$ for any open set $U \in \mathcal{U}$. For any $x \in \mathcal{B}$, we denote $\mathcal{F}_x := \pi^{-1}(x)$ and call it the **fibre** over $x$.*

**Definition 2 (Principal Bundle)** *Let $\mathcal{M}$ be a fibre bundle, and $\mathcal{G}$ a Lie group. We call $\mathcal{M}$ a **principal $\mathcal{G}$-bundle** if (1) $\mathcal{M}$ is a fibre bundle, (2) $\mathcal{M}$ admits a right action of $\mathcal{G}$ that preserves the fibres of $\mathcal{M}$, in the sense that for any $m \in \mathcal{M}$ we have $\pi(m) = \pi(g \cdot m)$, and (3) For any two points $p, q \in \mathcal{M}$ on the same fibre of $\mathcal{M}$, there exists a group element $g \in \mathcal{G}$ satisfying $p \cdot g = q$.*

If $\mathcal{M}$ is a principal $\mathcal{G}$-bundle over $\mathcal{B}$, any representation $\rho$ of $\mathcal{G}$ on a vector space $V$ induces an **associated vector bundle** over $\mathcal{B}$ with typical fibre $V$, denoted as $\mathcal{M} \times_\rho V$, defined as a quotient space $\mathcal{M} \times_\rho V := \mathcal{M} \times V / \sim$ where the equivalence relation is defined by $(m \cdot g, v) \sim (m, \rho(g)v)$ for all $m \in \mathcal{M}$, $g \in \mathcal{G}$, and $v \in V$. This construction gives rise to as many different associated vector bundles as the number of distinct representations of the Lie group $\mathcal{G}$. This allows us to study the $\mathcal{G}$-manifold $\mathcal{M}$, as a principal $\mathcal{G}$-bundle, through tools developed for learning an unknown manifold from attached vector bundle structures, such as *vector diffusion maps* (VDM) [57, 58]. We consider each of these associated vector bundles as a distinct "view" towards the unknown data manifold $\mathcal{M}$, as the representations inducing these vector bundles are different. In the rest of this paper, we will illustrate with several examples how to design learning and inference algorithms that exploit the inherent consistency in these associated vector bundles by representation theoretic machinery. Unlike the co-training setting where the consistency is induced from the labelled samples onto the unlabelled samples, in our unsupervised setting no labelled training data is provided and the consistency is induced purely from the geometry of the $\mathcal{G}$-manifold.

## 4  Methods

**Problem Setup:** Given a collection of $n$ data points $\{x_1, \ldots, x_n\} \subset \mathbb{R}^l$, we assume they lie on or close to a low dimensional smooth manifold $\mathcal{M}$ of intrinsic dimension $d \ll l$, and that $\mathcal{M}$ is a $\mathcal{G}$-*manifold* admitting the structure of a principal $\mathcal{G}$-bundle with a compact Lie group $\mathcal{G}$. The data space $\mathcal{M}$ is closed under the action of $\mathcal{G}$. That is, $g \cdot x \in \mathcal{M}$ for all group transformations $g \in \mathcal{G}$ and

data points $x \in \mathcal{M}$, where '·' denotes the group action. As an example, in a cryo-EM image dataset each image is a projection of a macromolecule with a random orientation, therefore $\mathcal{M} = \mathrm{SO}(3)$, which is the 3-D rotation group, $\mathcal{G} = \mathrm{SO}(2)$ which is the in-plane rotation of images. The $\mathcal{G}$-*invariant distance* $d_{ij}$ between two data points $x_i$ and $x_j$ is defined as

$$d_{ij} = \min_{g \in \mathcal{G}} \|x_i - g \cdot x_j\|, \quad \text{and} \quad g_{ij} = \arg\min_{g \in \mathcal{G}} \|x_i - g \cdot x_j\|. \tag{2}$$

where $\| \cdot \|$ is the Euclidean distance on the ambient space $\mathbb{R}^l$ and $g_{ij}$ is the associated alignment which is assumed to be unique. Then we build an undirected graph $G = (V, E)$ whose nodes are represented by data points, edge connection is given based on $d_{ij}$ using the $\epsilon$-neighborhood criterion, i.e. $(i, j) \in E$ iff $d_{ij} <= \epsilon$, or $\kappa$-nearest neighbor criterion, i.e. $(i, j) \in E$ iff $j$ is one of the $\kappa$ nearest neighbors of $i$. The edge weights $w_{ij}$ are defined using a kernel function on $d_{ij}$ as $w_{ij} = K_\sigma(d_{ij})$. The resulting graph $G$ is defined on the quotient space $\mathcal{B} := \mathcal{M}/\mathcal{G}$ and is invariant to the group transformations $g_{ij}$ within data points, e.g. for the viewing angles of cryo-EM images $\mathcal{B} = \mathrm{SO}(3)/\mathrm{SO}(2) = \mathrm{S}^2$. In a noiseless world, $G$ should be a neighborhood graph which only connects data points on $\mathcal{B}$ with small $d_{ij}$. However, in many applications, noise in the observational data severely degrades the estimations of $\mathcal{G}$-invariant distances $d_{ij}$ and optimal alignments $g_{ij}$. This leads to errors in the edge connection of $G$, which connect distant data points on $\mathcal{B}$ where their underlying geodesic distances are large.

Given the noisy graph, we consider the problem of *removing the wrong connections* and *recovering the underlying clean graph structure on $\mathcal{B}$*, especially under high level of noise. We propose a robust, unsupervised co-learning framework for addressing this, it has two steps which first builds a series of adjacency matrices with different irreps and filters the original noisy graph as denoising, further it checks the affinity between node pairs for identifying true neighbors in the clean graph. The main intuition is to *systematically explores the consistency* of the group transformation of the principal bundles across all irreps of $\mathcal{G}$, results in a robustness measurement of the affinity (see Fig. 1).

**Weight Matrices Using Irreps:** We start from building a series of weight matrices using multiple irreps of the compact Lie group $\mathcal{G}$. Given the graph $G = (V, E)$ with $n$ nodes and the group transformations $g \in \mathcal{G}$, we assign weight on each edge $(i, j) \in E$ by taking into account both the scalar edge connection weight $w_{ij}$ and the associated alignment $g_{ij}$ using unitary irreps $\rho_k$ for $k = 1, \ldots, k_{\max}$. The resulting graph can be described by a set of weight matrices $W_k$:

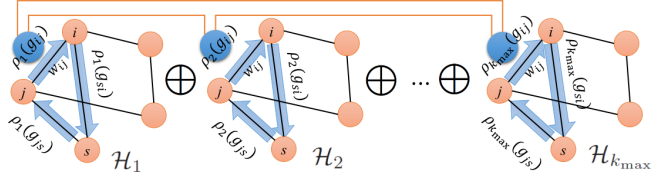

Figure 1: Illustration of our co-learning paradigm: Given a graph with irreps $\rho_k$ for $k = 1, \ldots, k_{\max}$, we identify neighbors by investigating the following consistencies: Within each graph of a single irrep $\rho_k$, if nodes $i, j$ and $s$ are neighbors, the cycle consistency of the group transformation holds: $\rho_k(g_{js})\rho_k(g_{si})\rho_k(g_{ij}) \approx I_{d_k \times d_k}$; Across different irreps, if $i, j$ are neighbors, the transformation $g_{ij}$ should be consistent algebraically (along the orange lines connecting the blue dots).

$$W_k(i, j) = \begin{cases} w_{ij}\rho_k(g_{ij}) & (i, j) \in E \\ 0 & \text{otherwise} \end{cases} \tag{3}$$

where $w_{ij} = w_{ji}$ and $\rho_k(g_{ji}) = \rho_k^*(g_{ij})$ for all $(i, j) \in E$. Recall the unitary irrep $\rho_k(g_{ij}) \in \mathbb{C}^{d_k \times d_k}$ is a $d_k \times d_k$ matrix, therefore $W_k$ is a block matrix with $n \times n$ blocks of size $d_k \times d_k$. In particular, the corresponding degree matrix $D_k$ is also a block diagonal matrix with the $(i, i)$-block $D_k(i, i)$ as:

$$D_k(i, i) = \deg(i)I_{d_k \times d_k}, \quad \deg(i) := \sum_{j:(i,j) \in E} w_{ij}. \tag{4}$$

The Hilbert space $\mathcal{H}$, as a unitary representation of the compact Lie group $\mathcal{G}$, admits an isotypic decomposition $\mathcal{H} = \bigoplus \mathcal{H}_k$, where a function $f$ is in $\mathcal{H}_k$ if and only if $f(xg) = g^k f(x)$. Then for each irrep $\rho_k$, we construct a normalized matrix $A_k = D_k^{-1} W_k$, which is an *averaging operator* for vector fields in $\mathcal{H}_k$. That is, for any vector $z_k \in \mathcal{H}_k$:

$$(A_k z_k)(i) = \frac{1}{\deg(i)} \sum_{j:(i,j) \in E} w_{ij}\rho_k(g_{ij})z_k(j). \tag{5}$$

Notice that $A_k$ is similar to a Hermitian matrix $\widetilde{A}_k$ as:

$$\widetilde{A}_k = D_k^{1/2} A_k D_k^{-1/2} = D_k^{-1/2} W_k D_k^{-1/2} \tag{6}$$

---

**Algorithm 1:** Weight Matrices Filtering

---

**Input:** Initial graph $G = (V, E)$ with $n$ nodes, for each $(i, j) \in E$ the scalar weight $w_{ij}$ and alignment $g_{ij}$, maximum frequency $k_{\max}$, cutoff parameter $m_k$ for $k = 1, \ldots, k_{\max}$, and spectral filter $\eta_t$

**Output:** The filtered weight matrices $\widetilde{W}_{k,t}$ for $k = 1, \ldots, k_{\max}$

1 **for** $k = 1, \ldots, k_{\max}$ **do**

2     Construct the block weight matrix $W_k$ of size $nd_k \times nd_k$ in (3) and the normalized symmetric matrix $\widetilde{A}_k$ in (6)

3     Compute the largest $m_k d_k$ eigenvalues $\lambda_1^{(k)} \geq \lambda_2^{(k)}, \geq, \ldots, \geq \lambda_{m_k d_k}^{(k)}$ of $\widetilde{A}_k$ and the corresponding eigenvectors $\{u_l^{(k)}\}_{l=1}^{m_k d_k}$

4     **for** $i = 1, \ldots, n$ **do**

5        Construct the $\mathcal{G}$-equivariant mapping, $\psi_t^{(k)} : i \mapsto \left[ \eta_t(\lambda_1)^{1/2} u_1^{(k)}(i), \ldots, \eta_t(\lambda_{m_k})^{1/2} u_{m_k d_k}^{(k)}(i) \right]$

6        (Optional) Compute the SVD of $\psi_t^{(k)}(i) = U\Sigma V^*$ and the normalized mapping $\widetilde{\psi}_t^{(k)}(i) = UV^*$.

7     **end**

8     Vertically concatenate $\widetilde{\psi}_t^{(k)}(i)$ or $\psi_t^{(k)}(i)$ to form the matrix $\Psi_t^{(k)}$ of size $nd_k \times m_k d_k$

9     Construct the filtered and normalized weight matrix $\widetilde{W}_{k,t} = \Psi_t^{(k)} \left( \Psi_t^{(k)} \right)^*$.

10 **end**

---

which has real eigenvalues and orthonormal eigenvectors $\{\lambda_l^{(k)}, u_l^{(k)}\}_{l=1}^{nd_k}$, and all the eigenvalues are within $[-1, 1]$. For simplicity, we assume data points are uniformly distributed on $\mathcal{B}$. If not, the normalization proposed in [17] can be applied to $W_k$. Now suppose there is a random walk on $G$ with a transition matrix $A_0$ and the trivial representations $\rho_0(g) = 1, \forall g \in \mathcal{G}$, then $A_0^{2t}(i, j)$ is the transition probability from $i$ to $j$ with $2t$ steps. Due to the usage of $\rho_0(g_{ij})$, $A_0^{2t}(i, j)$ not only takes into account the connectivity between the nodes $i$ and $j$, but also checks the consistency of transformations along all length-$2t$ paths between $i$ and $j$. Generally, in other cases when $k \geq 1$, $A_k^{2t}(i, j)$ is a sub-block matrix which still encodes such consistencies. Intuitively if $i, j$ are true neighbors on $G$, their transformations should be in agreement and we expect $\|A_k^{2t}(i, j)\|_{\mathrm{HS}}^2$ or $\|\widetilde{A}_k^{2t}(i, j)\|_{\mathrm{HS}}^2$ to be large, where $\| \cdot \|_{\mathrm{HS}}$ is the Hilbert-Schmidt norm. Previously, *vector diffusion maps* (VDM) [57, 58] considers $k = 1$ and defines the pairwise affinity as $\|\widetilde{A}_1^{2t}(i, j)\|_{\mathrm{HS}}^2$.

**Weight Matrices Filtering:** For denoising the graph, we generalize the VDM framework by first computing the filtered and normalized weight matrix $\widetilde{W}_{k,t} = \eta_t(\widetilde{A}_k)$ for all irreps $\rho_k$'s, where $\eta_t(\cdot)$ denotes a spectral filter acting on the eigenvalues, for example $\eta_t(\lambda) = \lambda^{2t}$ as VDM. Moreover, since the small eigenvalues of $\widetilde{A}_k$ are more sensitive to noise, a truncation is applied by only keeping the top $m_k d_k$ eigenvalues and eigenvectors. Specifically, we equally divide $u_l^{(k)}$ of length $nd_k$ into $n$ blocks and denote the $i$th block as $u_l^{(k)}(i)$. In this way, we define a $\mathcal{G}$-*equivariant mapping* as:

$$\psi_t^{(k)} : i \mapsto \left[ \eta_t(\lambda_1)^{1/2} u_1^{(k)}(i), \ldots, \eta_t(\lambda_{m_k d_k})^{1/2} u_{m_k d_k}^{(k)}(i) \right] \in \mathbb{C}^{d_k \times m_k d_k}, \quad i = 1, 2, \ldots, n. \quad (7)$$

It can be further normalized to ensure the diagonal blocks of $\widetilde{W}_{k,t}$ are identity matrices, i.e. $\widetilde{W}_{k,t}(i, i) = I_{d_k \times d_k}$ for all nodes $i$. The steps for weight matrices filtering are detailed in Alg. 1. The resulting denoised $\widetilde{W}_{k,t}$ is then used for building our affinity measures.

**Optimal Alignment Affinity:** At each irrep $\rho_k$, the filtered $\widetilde{W}_{k,t}$ involves the transformation consistency of the graph represented by $W_k$ and has its own ability to measure the affinity. Then similar to the unsupervised multi-view learning approach, it is advantageous to boost this by coupling the information under different irreps and to achieve a more accurate measurement (see Fig. 1). Furthermore, notice that if $i$ and $j$ are true neighbors, for each irrep $\rho_k$ the block $\widetilde{W}_{k,t}(i, j)$ should encode the same amount of associated alignment $g_{ij}$. Therefore, by applying the algebraic relation among $\widetilde{W}_{k,t}$ across all irreps, we define the *optimal alignment affinity* according to the generalized Fourier transform in (1) and the definition of the weight matrices in (3):

$$S_t^{\mathrm{OA}}(i, j) := \max_{g \in \mathcal{G}} \frac{1}{k_{\max}} \left| \sum_{k=1}^{k_{\max}} \mathrm{Tr} \left[ \widetilde{W}_{k,t}(i, j) \rho_k^*(g) \right] \right|, \quad (8)$$

which can be evaluated using generalized FFTs [39]. Here both the cycle consistency within each graph and the algebraic relation across different irreps in Fig. 1 are considered.

**Power Spectrum Affinity:** Searching for the optimal alignment among all transformations as above could be computationally challenging and extremely time consuming. Therefore, invariant features can be used to speed up the computation. First we consider the *power spectrum*, which is the Fourier transform of the auto-correlation defined as $P_f(k) = F_k F_k^*$ according to the convolution theorem. It is transformation invariant since under the right action of $g \in \mathcal{G}$, the Fourier coefficients $F_k \rightarrow F_k \rho_k(g)$ and $P_{f \cdot g}(k) = F_k \rho_k(g) \rho_k(g)^* F_k^* = P_f(k)$. Hence, for each $k$ we compute the power spectrum $P_k$ of $\widetilde{W}_{k,t}$ and combine them as the *power spectrum affinity*:

$$S_t^{\text{power spec}}(i,j) = \frac{1}{k_{\max}} \sum_{k=1}^{k_{\max}} \text{Tr}\left[P_k(i,j)\right], \quad \text{with } P_k(i,j) = \widetilde{W}_{k,t}(i,j)\widetilde{W}_{k,t}(i,j)^*, \quad (9)$$

which does not require the search of optimal alignment and is thus computationally efficient. Recently, multi-frequency vector diffusion maps (MFVDM) [20] considers $\mathcal{G} = \text{SO}(2)$ and sums the power spectrum at different irreps as their affinity. Here, we extend it to a general compact Lie group.

**Bispectrum Affinity:** Although, the power spectrum affinity combines the information at different irreps, it does not couple them and loses the *relative phase information*, i.e. the transformation across different irreps $\rho_k$ (see Fig. 1). Consequently, the affinity might be inaccurate under high level of noise. In order to systematically impose the algebraic consistency without solving the optimization problem in (8), we consider another invariant feature called *bispectrum*, which is the Fourier transform of the triple correlation and has been used in several fields [32, 27, 72, 31]. Formally, let us consider two unitary irreps $\rho_{k_1}$ and $\rho_{k_2}$ on finite dimensional vector spaces $\mathcal{H}_{k_1}$ and $\mathcal{H}_{k_2}$ of the compact Lie group $\mathcal{G}$. There is a unique decomposition of $\rho_{k_1} \bigotimes \rho_{k_2}$ into a set of unitary irreps $\rho_k$, $k \in \mathbb{N}$, where $\bigotimes$ is the Kronecker product of matrices, and we use $\bigoplus$ to denote direct sum. There exists $\mathcal{G}$-equivariant maps from $\mathcal{H}_{k_1} \bigotimes \mathcal{H}_{k_2} \rightarrow \bigoplus \mathcal{H}_k$, called generalized *Clebsch–Gordan* coefficients $C_{k_1,k_2}$ for $\mathcal{G}$, which satisfies:

$$\rho_{k_1}(g) \bigotimes \rho_{k_2}(g) = C_{k_1,k_2} \left[\bigoplus_{k \in k_1 \bigotimes k_2} \rho_k(g)\right] C_{k_1,k_2}^*. \quad (10)$$

Using (10) and the fact that $C_{k_1,k_2}$ and $\rho_k$'s are unitary matrices, we have

$$\left[\rho_{k_1}(g) \bigotimes \rho_{k_2}(g)\right] C_{k_1,k_2} \left[\bigoplus_{k \in k_1 \bigotimes k_2} \rho_k^*(g)\right] C_{k_1,k_2}^* = I_{d_{k_1} d_{k_2} \times d_{k_1} d_{k_2}}. \quad (11)$$

Particularly, the triple correlation of a function $f$ on $\mathcal{G}$ can be defined as $a_{3,f}(g_1,g_2) = \int_{\mathcal{G}} f^*(g)f(gg_1)f(gg_2)d\mu_g$. Then the bispectrum is defined as the Fourier transform of $a_{3,f}$ as

$$B_f(k_1,k_2) = \left[F_{k_1} \bigotimes F_{k_2}\right] C_{k_1,k_2} \left[\bigoplus_{k \in k_1 \bigotimes k_2} F_k^*\right] C_{k_1,k_2}^*. \quad (12)$$

Under the action of $g$, we have the following properties of the Fourier coefficients of $f$: (1) $F_k \rightarrow F_k \rho_k(g)$, and (2) $F_{k_1} \bigotimes F_{k_2} \rightarrow (F_{k_1} \rho_{k_1}(g)) \bigotimes (F_{k_1} \rho_{k_1}(g)) = (F_{k_1} \bigotimes F_{k_1})(\rho_{k_1}(g) \bigotimes \rho_{k_2}(g))$. Therefore, $B_f$ is $\mathcal{G}$-invariant according to (11) and (12). By combining the bispectrum at different $k_1$ and $k_2$, we establish the *bispectrum affinity* as,

$$S_t^{\text{bispec}}(i,j) = \frac{1}{k_{\max}^2} \left|\sum_{k_1=1}^{k_{\max}} \sum_{k_2=1}^{k_{\max}} \text{Tr}\left[B_{k_1,k_2,t}(i,j)\right]\right|, \quad \text{with} \quad (13)$$

$$B_{k_1,k_2,t}(i,j) = \left[\widetilde{W}_{k_1,t}(i,j) \bigotimes \widetilde{W}_{k_2,t}(i,j)\right] C_{k_1,k_2} \left[\bigoplus_{k \in k_1 \bigotimes k_2} \widetilde{W}_{k,t}^*(i,j)\right] C_{k_1,k_2}^*. \quad (14)$$

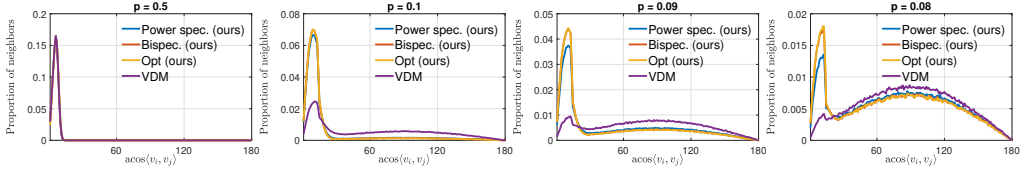

Figure 2: Histograms of $\arccos \langle v_i, v_j \rangle$ between estimated nearest neighbors on $\mathcal{M} = \mathrm{SO}(3)$, $\mathcal{G} = \mathrm{SO}(2)$ and $\mathcal{B} = \mathrm{S}^2$ with different SNRs. The clean histogram should peak at small angles. The lines of the bispectrum and the optimal alignment affinities almost overlap in all these plots. We set $k_{\max} = 6$, $m_k = 10$ for all $k$'s and $t = 1$.

If the transformations are consistent across different $k$'s, the trace of $B_{k_1,k_2,t}$ in (14) should be large. Therefore, this affinity not only takes into account the consistency of the transformation at each irrep, but also explores the algebraic consistency across different irreps.

**Higher Order Invariant Moments:** The power spectrum and bispectrum are second-order and third-order cumulants, certainly it is possible to design affinities by using higher order invariant features. For example, we can define the order-$d + 1$ $\mathcal{G}$-invariant features as: $M_{k_1,\ldots,k_d} = [F_{k_1} \bigotimes \cdots \bigotimes F_{k_d}] C_{k_1,\ldots,k_d} \left[ \bigoplus_{k \in k_1 \otimes \cdots \otimes k_d} F_k^* \right] C_{k_1,\ldots,k_d}^*$, where $C_{k_1,\ldots,k_d}$ is the extension of the Clebsch–Gordan coefficients. However, using higher order spectra dramatically increases the computational complexity. In practice, the bispectrum is sufficient to check the consistency of the group transformations between nodes and across all irreps.

**Computational Complexity:** Filtering the normalized weight matrix involves computing the top $m_k d_k$ eigenvectors of the sparse Hermitian matrices $A_k$, for $k = 1, \ldots, k_{\max}$, which can be efficiently evaluated using block Lanczos method [51], and its cost is $O(nm_k d_k^2 (m_k + l_k))$, where $l_k$ is the average number of non-zero elements in each row of $\widetilde{A}_k$. We compute the spectral decomposition for different $k$'s in parallel. Computing the power spectrum invariant affinity for all pairs takes $O(n^2 \sum_{k=1}^{k_{\max}} d_k^2)$ flops. The computational complexity of evaluating the bispectrum invariant affinity is $O(n^2 (\sum_{k_1=0}^{k_{\max}} \sum_{k_2=0}^{k_{\max}} d_{k_1}^2 d_{k_2}^2))$. For the optimal alignment affinity, the computational complexity depends on the cost of optimal alignment search $C_a$ and the total cost is $O(n^2 C_a)$. For certain group structures, where FFTs are developed, the optimal alignment affinity can be efficiently and accurately approximated. However, $C_a$ is still larger than the computation cost of invariants.

**Examples with $\mathcal{G} = \mathbf{SO(2)}$ and $\mathbf{SO(3)}$:** If the group transformation is 2-D in-plane rotation, i.e. $\mathcal{G} = \mathrm{SO}(2)$, the unitary irreps will be $\rho_k(\alpha) = e^{ik\alpha}$, where $\alpha \in (0, 2\pi]$ is the rotation angle. The dimensions of the irreps are $d_k = 1$, and $k_1 \bigotimes k_2 = k_1 + k_2$. The generalized Clebsch–Gordan coefficients is 1 for all $(k_1, k_2)$ pairs. If $\mathcal{G}$ is the 3-D rotation group, i.e. $\mathcal{G} = \mathrm{SO}(3)$, the unitary irreps are the Wigner D-matrices for $\omega \in \mathrm{SO}(3)$ [68]. The dimensions of the irreps are $d_k = 2k + 1$, and $k_1 \bigotimes k_2 = \{|k_1 - k_2|, \ldots, k_1 + k_2\}$. The Clebsch–Gordan coefficients for all $(k_1, k_2)$ pairs can be numerically precomputed [26]. These two classical examples are frequently used in the real world and are investigated in our experiments.

## 5 Experiments

We evaluate our paradigm through three examples: (1) Nearest neighbor (In brevity: **NN**) search on 2-sphere $\mathrm{S}^2$ with $\mathcal{G} = \mathrm{SO}(2)$; (2) nearest viewing angle search for cryo-EM images; (3) spectral clustering with $\mathcal{G} = \mathrm{SO}(2)$ or $\mathcal{G} = \mathrm{SO}(3)$ transformation. We compare with the baseline vector diffusion maps (VDM) [57]. In particular, since the greatest advantage of our paradigm is the robustness to noise, we demonstrate this through datasets contaminated by extremely high level of noise. The setting of hyper parameters, e.g. $k_{\max}$ and $m_k$, are shown in the captions, we point out that our algorithm is not sensitive to the choice of parameters. The experiments are conducted in MATLAB on a computer with Intel i7 7th generation quad core CPU.

**NN Search for $\mathcal{M} = \mathbf{SO(3)}$, $\mathcal{G} = \mathbf{SO(2)}$, $\mathcal{B} = \mathbf{S^2}$:** We simulate $n = 10^4$ points uniformly distributed over $\mathcal{M} = \mathrm{SO}(3)$ according to the Haar measure. Each point can be represented by a $3 \times 3$ orthogonal matrix $R = [R^1, R^2, R^3]$, whose determinant is equal to 1. Then the vector $v = R^3$ can be realized as a point on the unit 2-sphere (i.e. $\mathcal{B} = \mathrm{S}^2$). The first two columns $R^1$ and $R^2$ spans

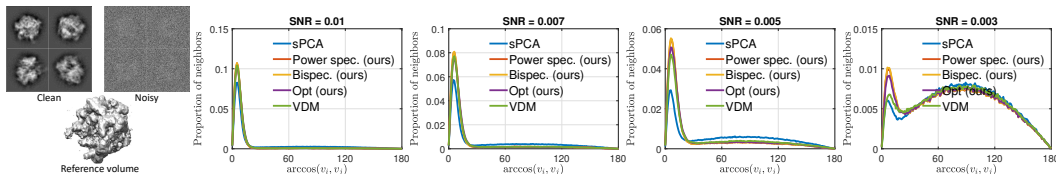

Figure 3: Nearest viewing angle search for cryo-EM images. *Left*: clean, noisy (SNR = 0.01) projections image samples, and reference volume of 70s ribosome; *Right*: Histograms of $\arccos \langle v_i, v_j \rangle$ between estimated nearest neighbors. sPCA is the initial noisy input of our graph structure. The lines of power spectrum and bispectrum almost overlap in all these plots. We set $k_{\max} = 20$, $m_k = 20$ for all $k$'s and $t = 1$.

the tangent plane of the sphere at $v$. Given two points $i$ and $j$, there exists a rotation angle $\alpha_{ij}$ that optimally aligns the tangent bundles $[R_j^1, R_j^2]$ to $[R_i^1, R_i^2]$ as in (2). Therefore, the manifold $\mathcal{M}$ is a $\mathcal{G}$-manifold with $\mathcal{G} = \mathrm{SO}(2)$. Then we build a clean neighborhood graph $G = (V, E)$ by connecting nodes with $\langle v_i, v_j \rangle \geq 0.97$, and add noise following a *random rewiring model* [59]. With probability $p$, we keep the existing edge $(i, j) \in E$. With probability $1 - p$, we remove it and link $i$ to another vertex drawn uniformly at random from the remaining vertices that are not already connected to $i$. For those rewired edges, their alignments are uniformly distributed over $[0, 2\pi]$ according to the Haar measure. In this way, the probability $p$ controls the signal to noise ratio (SNR) where $p = 1$ indicates the clean case, while $p = 0$ is fully random. For each node, we identify its 50 NNs based on the three proposed affinities and the affinity in VDM. In Fig. 2 we plot the histogram of $\arccos \langle v_i, v_j \rangle$ of identified NNs under different SNRs. When $p = 0.08$ to $p = 0.1$ (over 90% edges are corrupted), bispectrum and optimal alignment achieve similar result and outperform power spectrum and VDM. This indicates our proposed affinities are able to recover the underlying clean graph, even at an extremely high noise level.

**Nearest Viewing Angle Search for Cryo-EM Images:** One important application of the NN search above is in *cryo-EM image analysis*. Given a series of projection images of a macromolecule with unknown random orientations and extremely low SNR (see Fig. 3), we aim to identify images with similar projection directions and perform local rotational alignment, then image SNR can be boosted by averaging the aligned images. Therefore, each projection image can be viewed as a point lying on the 2-sphere (i.e. $\mathcal{B} = \mathrm{S}^2$), and the group transformation is the in-plane rotation of an image (i.e., $\mathcal{G} = \mathrm{SO}(2)$).

In our experiments, we simulate $n = 10^4$ projection images from a 3D electron density map of the 70S ribosome, the orientations of all projections are uniformly distributed over $\mathrm{SO}(3)$ and the images are contaminated by additive white Gaussian noise (see Fig. 3 for noisy samples). As preprocessing, we build the initial graph $G$ by using fast steerable PCA (sPCA) [71] and rotationally invariant features [72] to initially identify the images of similar views and the corresponding in-plane rotational alignments. Similar to the example above, we compute the affinities for NNs identification. In Fig. 3, we display the histograms of $\arccos \langle v_i, v_j \rangle$ of identified NNs under different SNRs. Result shows that all proposed affinities outperform VDM. The power spectrum and the bispectrum affinities achieve similar result, and outperform the optimal alignment affinity. This result is different from the previous example with the random rewiring model on $\mathrm{S}^2$. This is because those two examples have different noise model, the random rewiring model has independent noise on edges, whereas the examples using cryo-EM images have independent noise on nodes with dependent noise on edges.

**Spectral Clustering with $\mathrm{SO}(2)$ or $\mathrm{SO}(3)$ Transformations:** We apply our framework to spectral clustering. In particular, we assume there exists a group transformation $g_{ij} \in \mathcal{G}$ in addition to the scalar weight $w_{ij}$ between members (nodes) in a network. Formally, given $n$ data points with $K$ equal sized clusters, for each point $i$, we uniformly assign an in-plane rotational angle $\alpha_i \in [0, 2\pi)$, or a 3-D rotation $\omega_i \in \mathrm{SO}(3)$. Then the optimal alignment is $\alpha_{ij} = \alpha_i - \alpha_j$, or $\omega_{ij} = \omega_i \omega_j^{-1}$. We build the clean graph by fully connecting nodes within each cluster. The noisy graph is then built following the random rewiring model with a rewiring probability $p$. We perform clustering by using our proposed affinities as the input of spectral clustering, and compare with the traditional spectral clustering [45, 65] which only takes into account the scalar edge connection, and VDM [57], which defines affinity based on the transformation consistency at a single representation. In Tab. 1, we use *Rand index* [48] to measure the performance (larger value is better). Our three affinities achieve similar accuracy and they outperform the traditional spectral clustering (scalar) and VDM. The results reported in Tab. 1 are evaluated over 50 trials for $\mathrm{SO}(2)$ and 10 trials for $\mathrm{SO}(3)$ respectively.

Table 1: Rand index (larger value is better) of spectral clustering results with SO(2) or SO(3) group transformation. We set the number of clusters *Left*: $K = 2$ and *right*: $K = 10$. For $K = 10$ and SO(3) case, each cluster has 25 points, otherwise each cluster has 50 points. We set $m_k = K$, $k_{\max} = 10$ and $t = 1$ for all cases.

| $\mathcal{G}$ | method | $K = 2$ clusters | | | $K = 10$ clusters | | |
|---|---|---|---|---|---|---|---|
| | | $p = 0.16$ | $p = 0.20$ | $p = 0.25$ | $p = 0.16$ | $p = 0.20$ | $p = 0.25$ |
| SO(2) | Scalar | $0.569 \pm 0.069$ | $0.705 \pm 0.092$ | $0.837 \pm 0.059$ | $0.868 \pm 0.010$ | $0.948 \pm 0.015$ | $0.981 \pm 0.013$ |
| | VDM | $0.526 \pm 0.036$ | $0.644 \pm 0.076$ | $0.857 \pm 0.057$ | $0.892 \pm 0.010$ | $0.963 \pm 0.011$ | $0.994 \pm 0.008$ |
| | **Power spec. (ours)** | $0.670 \pm 0.065$ | $0.899 \pm 0.051$ | $0.981 \pm 0.021$ | $0.975 \pm 0.010$ | $0.991 \pm 0.011$ | $0.998 \pm 0.006$ |
| | **Opt (ours)** | $\mathbf{0.687 \pm 0.011}$ | $\mathbf{0.912 \pm 0.009}$ | $\mathbf{0.986 \pm 0.007}$ | $\mathbf{0.976 \pm 0.012}$ | $0.994 \pm 0.008$ | $0.997 \pm 0.005$ |
| | **Bispec. (ours)** | $0.664 \pm 0.073$ | $0.901 \pm 0.062$ | $0.983 \pm 0.019$ | $0.967 \pm 0.014$ | $\mathbf{0.997 \pm 0.003}$ | $\mathbf{1 \pm 0}$ |
| SO(3) | Scalar | $0.572 \pm 0.061$ | $0.666 \pm 0.095$ | $0.862 \pm 0.056$ | $0.838 \pm 0.003$ | $0.838 \pm 0.007$ | $0.909 \pm 0.019$ |
| | VDM | $0.600 \pm 0.048$ | $0.840 \pm 0.056$ | $0.974 \pm 0.023$ | $0.850 \pm 0.011$ | $0.919 \pm 0.013$ | $0.965 \pm 0.014$ |
| | **Power spec. (ours)** | $\mathbf{0.921 \pm 0.038}$ | $0.986 \pm 0.016$ | $\mathbf{1 \pm 0}$ | $\mathbf{0.874 \pm 0.011}$ | $0.939 \pm 0.011$ | $\mathbf{0.981 \pm 0.017}$ |
| | **Bispec. (ours)** | $0.911 \pm 0.043$ | $\mathbf{0.990 \pm 0.010}$ | $\mathbf{1 \pm 0}$ | $0.869 \pm 0.012$ | $\mathbf{0.943 \pm 0.009}$ | $0.979 \pm 0.011$ |

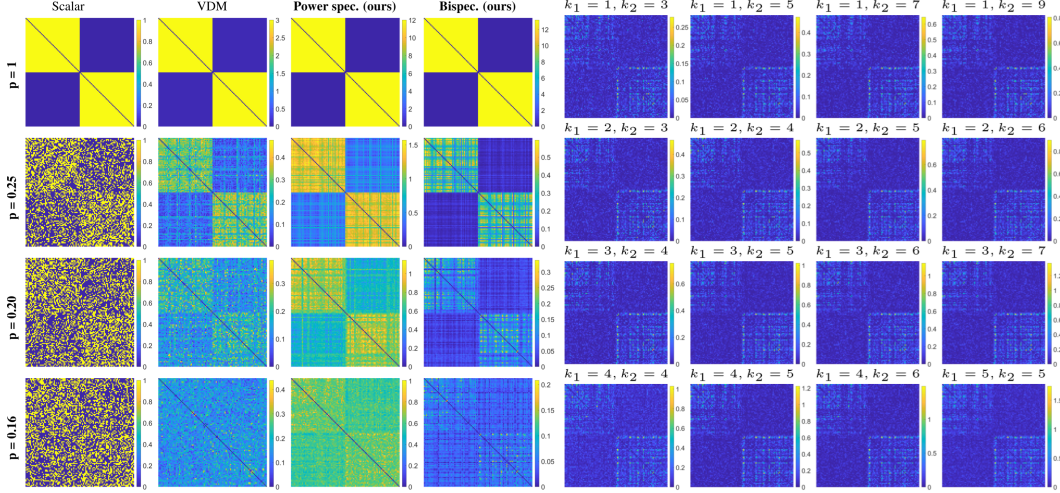

(a) Affinity matrix of different methods      (b) Bispectrum affinity component $|\mathrm{Tr}(B_{k_1,k_2,t}(i,j))|$

Figure 4: Spectral clustering for $K = 2$ clusters with SO(3) group transformation. The underlying clean graph is corrupted according to the random rewiring model. *Left*: Plot of the affinity matrix by different approaches. The clusters are of equal size and form two diagonal blocks in the clean affinity matrix (see the *scalar* column at $p = 1$). Here we do not include the affinity of each node with itself and the diagonal entries are 0; *Right*: Plot of the bispecturm affinity $|\mathrm{Tr}[B_{k_1,k_2,t}(i,j)]|$ at different $k_1, k_2$, $p = 0.16$.

For a better understanding, we visualize the $n \times n$ affinity matrices by different approaches as shown in Fig. 4a at $K = 2$ and $\mathcal{G} = \mathrm{SO}(3)$. We observe that at high noise levels, such as $p = 0.16$ or $0.2$, the underlying 2-cluster structure is visually easier to be identified through our proposed affinities. In particular, as the bispectrum affinity in (13) is the combination of the bispectrum coefficients $B_f(k_1, k_2)$, Fig. 4b shows the component $|\mathrm{Tr}[B_{k_1,k_2,t}(i,j)]|$ at different $k_1, k_2$. Visually, the 2-cluster structure appears in each $(k_1, k_2)$ component with some variations across different components. Combining those information together results in a more robust classifier.

## 6  Conclusion

In this paper, we propose a novel mathematical and computational framework for unsupervised co-learning on $\mathcal{G}$-manifolds across multiple unitary irreps for robust nearest neighbor search and spectral clustering. We have a two stage algorithm: At the first stage, the graph adjacency matrices are individually denoised through spectral filtering. This step uses the local cycle consistency of the group transformation; The second stage checks the algebraic consistency over different irreps and we propose three different ways to combine the information across all irreps. Using invariant moments bypasses the pairwise alignment and is computationally more efficient than the affinity based on the optimal alignment search. Experimental results show the efficacy of the framework compared to the state-of-the-art methods, which do not take into account of the transformation group or only use a single representation.

**Acknowledgement:** This work is supported in part by the National Science Foundation DMS-185479 and DMS-1854831.

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
