[Supplementary Material]

# Supplementary Materials: Unsupervised Co-Learning on $\mathcal{G}$-Manifolds Across Irreducible Representations

**Yifeng Fan**[1]    **Tingran Gao**[2]    **Zhizhen Zhao**[1]
[1]University of Illinois at Urbana-Champaign    [2]University of Chicago
{yifengf2, zhizhenz}@illinois.edu    tingrangao@galton.uchicago.edu

## 1   Additional Results on Spectral Clustering

In the main paper, we visualize the $n \times n$ affinity matrices of $K = 2$ clusters with $\mathcal{G} = \mathrm{SO}(3)$ for spectral application, in the presence of edge noise. Here we provide another visualization of the affinity measures for the results in Table 1 of the main paper, with $K = 10$ and $\mathcal{G} = \mathrm{SO}(3)$. In Fig. 1, we show the affinity matrices using single frequency ($k = 1$) VDM, power spectrum, and bispectrum. The cutoff parameter $m_k$ and maximum frequency $k_{\max}$ are set as $m_k = K = 10$ and $k_{\max} = 10$. We observe similar patterns for the 2-cluster example (see Fig. 4a in the main paper). For noisy examples with $p = 0.20$ and $0.16$, the cluster structure is more easily identified through our proposed affinities compared to scalar edge weights used in the traditional spectral clustering [1], and frequency $k = 1$ VDM [2]. This again demonstrates the efficacy of our approach in estimating the cluster structures in the presence of large level of noise on edges.

## 2   Performance under Different Choices of Parameters

In this section, we include more numerical results to show the performance of our methods under different parameter settings and provide theoretical justification under a probabilistic model.

### 2.1   Nearest Neighbor Identification on Base Manifold

First, we analyze the spectral properties of the matrix $W_k$ based on the random rewiring model [3]. Starting from the underlying true graph, we perturb the graph in the following way: with probability $1 - p$, we remove the clean edge $(i, j) \in E$ and create a link between $i$ and some random vertex, drawn uniformly at random from the remaining vertices that are not connected to $i$. If the link between $i$ and $j$ is a rewired random link, then the associated group element $g_{ij}$ is distributed over $\mathcal{G}$ according to the Haar measure. The corresponding matrix $W_k$ is a random matrix under this model. If the distribution of $g$ is the Haar measure on $\mathcal{G}$, we have $\mathbb{E}\rho_k(g) = 0_{d_k \times d_k}$ for $k \neq 0$. Therefore, we get $\mathbb{E}W_k = pW_k^{\mathrm{clean}}$ for $k \neq 0$, where $W_k^{\mathrm{clean}}$ is the matrix with all links and group elements inferred correctly ($p = 1$). Thus the matrix $W_k$ can be decomposed as,

$$W_k = pW_k^{\mathrm{clean}} + R_k, \tag{1}$$

where $R_k$ is a Hermitian random matrix with random blocks. The upper triangular part of the matrix contains independent random blocks with finite moments (the elements of $R_k$ are all bounded). Thus we use $p$ to describe the signal-to-noise ratio of the observed graph. According to the matrix perturbation theory, the top eigenvectors of $W_k$ approximates the top eigenvectors of $W_k^{\mathrm{clean}}$ as long as the 2-norms of $R_k$ is not too large.

We numerically test the sensitivity of our methods to the choice of parameters in application to the nearest neighbor identification on base manifold. The set up of the experiments is similar to Section

Figure 1: Similarity measure for $K = 10$ clusters with $SO(3)$ group transformation. The underlying clean graph is corrupted according to the random rewiring model. We show the plot of the affinity matrix by different approaches. The clusters are of equal size and form 10 diagonal blocks in the clean affinity matrix (see the *scalar* column at $p = 1$). Here we do not include the affinity of each node with itself and the diagonal entries are 0.

5 in the main paper. We simulate $n = 10^4$ data points uniformly sampled from $\mathcal{M} = SO(3)$ and build the clean neighborhood graph on $\mathcal{B} = S^2$. The random rewiring perturbation is then applied to the clean graph and the nearest neighbors are identified based on the proposed affinities, with two varying parameters: cutoff parameter $m_k$ and maximum frequency $k_{\max}$. We evaluate by computing the proportion (in percentage) of all identified nearest neighbor pairs $(i, j)$'s whose $\langle v_i, v_j \rangle > 0.95$. The results are shown in Tab. 1 and Tab. 2. We have the following observation.

**Cutoff Parameter $m_k$:** Ideally, at each frequency $k$, the truncation cutoff $m_k$, that is, selecting the top $m_k d_k$ eigenvectors of matrix $\widetilde{A}_k = D_k^{-1/2} W_k D_k^{-1/2}$, should be set to include top eigenvectors that are not largely perturbed by noise and have nontrivial correlation with the eigenvectors of the clean matrix $\widetilde{A}_k^{\text{clean}}$. This value can vary between different frequencies. However, in practice, we set $m_k$ to be a moderate constant for all $k$'s. In all the trials, we set $k_{\max} = 10$. In Tab. 1, we observe that the accuracy is first improved when $m_k$ increases since more information is included. However, the accuracy degrades or gets saturated when $m_k$ is larger than a certain $p$ dependent value due to the effects of noise. This implies a moderate $m_k$ is needed for a trade-off between the useful information and the impact of noise.

**Maximum Frequency $k_{\max}$:** We test $k_{\max}$ from 2 to 100 and show the results in Tab. 2. We fix $m_k = 20$, when varying $k_{\max}$. In the extremely noisy cases, such as $p = 0.09$ and $0.10$, the results improve when $k_{\max}$ increases within the range of the values we test. When $p > 0.1$, the accuracy first increases but then degrades or gets saturated after a certain $p$ dependent value of $k_{\max}$. This indicates that the optimal choice of $k_{\max}$ depends on the noise level. Under this particular noise perturbation model, the higher the noise level is, the larger the $k_{\max}$ is needed. Also, since for all three proposed affinities the computation complexity greatly increase with a growing $k_{\max}$, it should be chosen that our computation budget can afford.

Table 1: The accuracy of nearest neighbor identification on base manifold with varying cutoff $m_k$ for $\mathcal{M} =$ SO(3), $\mathcal{G} =$ SO(2) and $\mathcal{B} = \mathrm{S}^2$. The maximum frequency is $k_{\max} = 10$ in all the experiments. We compute the proportion (in percentage) of identified neighbor pairs $(i,j)$'s whose $\langle v_i, v_j \rangle > 0.95$, at different signal-to-noise ratios $p$'s. For each method we highlight its best result in boldface.

| $p$ | method | Truncation cutoff $m_k$ | | | | | |
|---|---|---|---|---|---|---|---|
| | | 2 | 5 | 10 | 20 | 50 | 100 |
| 0.08 | VDM | 2.63 | 3.02 | 3.48 | 3.67 | 4.14 | **4.59** |
| | **Power spec. (ours)** | 2.91 | 4.71 | 5.93 | 7.05 | 9.16 | **12.30** |
| | **Opt (ours)** | 2.94 | 5.66 | 7.26 | 8.95 | 12.20 | **16.43** |
| | **Bispec. (ours)** | 2.88 | 5.53 | 7.24 | 8.70 | 11.91 | **16.23** |
| 0.09 | VDM | 2.82 | 4.60 | 8.05 | **9.46** | 9.25 | 9.13 |
| | **Power spec. (ours)** | 4.37 | 14.96 | 33.44 | **38.85** | 38.21 | 37.77 |
| | **Opt (ours)** | 5.64 | 22.65 | 45.70 | **51.59** | 50.93 | 49.77 |
| | **Bispec. (ours)** | 5.62 | 22.17 | 44.84 | **50.44** | 49.57 | 48.68 |
| 0.10 | VDM | 3.48 | 8.65 | 17.96 | **27.56** | 24.58 | 20.87 |
| | **Power spec. (ours)** | 8.29 | 38.22 | 68.09 | **83.04** | 78.92 | 73.56 |
| | **Opt (ours)** | 15.03 | 52.25 | 77.38 | **87.72** | 86.25 | 82.66 |
| | **Bispec. (ours)** | 14.95 | 51.19 | 76.57 | **87.33** | 85.56 | 81.90 |
| 0.5 | VDM | 57.04 | 98.48 | 99.99 | **100** | **100** | **100** |
| | **Power spec. (ours)** | 99.05 | **100** | **100** | **100** | **100** | **100** |
| | **Opt (ours)** | 99.60 | 99.99 | **100** | **100** | **100** | **100** |
| | **Bispec. (ours)** | 99.60 | **100** | **100** | **100** | **100** | **100** |

Table 2: The accuracy of nearest neighbor identification on base manifold with varying maximum frequency $k_{\max}$ for $\mathcal{M} =$ SO(3), $\mathcal{G} =$ SO(2) and $\mathcal{B} = \mathrm{S}^2$. The cutoff parameter is $m_k = 20$ in all the experiments. We compute the proportion (in percentage) of identified neighbor pairs $(i,j)$'s whose $\langle v_i, v_j \rangle > 0.95$, at different signal-to-noise ratios $p$'s. For each method we highlight its best result in boldface. Note that VDM only uses single frequency $k_{\max} = 1$.

| $p$ | method | Maximum frequency $k_{\max}$ | | | | |
|---|---|---|---|---|---|---|
| | | 2 | 5 | 10 | 20 | 50 |
| 0.08 | VDM | | | — 3.67 — | | |
| | **Power spec. (ours)** | 4.12 | 5.23 | 7.05 | 9.52 | **11.45** |
| | **Opt (ours)** | 4.06 | 5.39 | 8.95 | 16.59 | **29.17** |
| | **Bispec. (ours)** | 4.03 | 5.26 | 8.70 | 15.73 | **26.55** |
| 0.09 | VDM | | | — 9.46 — | | |
| | **Power spec. (ours)** | 13.47 | 25.49 | 38.85 | 52.02 | **55.40** |
| | **Opt (ours)** | 13.28 | 29.74 | 51.59 | 71.21 | **76.30** |
| | **Bispec. (ours)** | 13.03 | 28.85 | 50.44 | 70.18 | **77.26** |
| 0.10 | VDM | | | — 27.56 — | | |
| | **Power spec. (ours)** | 43.66 | 69.29 | 83.04 | 90.15 | **90.56** |
| | **Opt (ours)** | 43.42 | 73.69 | 87.72 | **93.01** | 92.07 |
| | **Bispec. (ours)** | 42.15 | 72.55 | 87.33 | 93.05 | **93.20** |
| 0.5 | VDM | | | — 100 — | | |
| | **Power spec. (ours)** | 100 | 100 | 100 | 100 | 100 |
| | **Opt (ours)** | 100 | 100 | 100 | 100 | 100 |
| | **Bispec. (ours)** | 100 | 100 | 100 | 100 | 100 |

## 2.2 Spectral Clustering

We check the performance of our methods in spectral clustering under different parameter settings. From the clean cluster graph, we apply the random rewiring perturbation as described above.

**Cutoff Parameter $m_k$:** In the clean case, the number of non-zero eigenvalues of the weight matrices $W_k$ is $d_k K$ for $K$ clusters. Therefore, each $W_k$ has a low-rank structure and so is the normalized Hermitian matrix $\widetilde{A}_k = D_k^{-1/2} W_k D_k^{-1/2}$. Then a truncation at top $d_k K$ eigenvectors (i.e. $m_k = K$) is enough for clustering. In the noise case, following the model in (1), we are still able to use the top $d_k K$ eigenvectors for clustering as long as the signal-to-noise ratio $p$ is not too small. Using less eigenvectors as $m_k < K$ will lead to loss of information and using $m_k > K$ will include spurious information from noise. We conduct the experiments with $K = 10$ clusters, where each cluster contains 50 points, and $\mathcal{G} =$ SO(2). We set $p = 0.16, 0.2, 0.25$ and $k_{\max} = 10$. We vary the cutoff $m_k$ from 2 to 100 and display the Rand indices of the clustering results from different methods in Tab. 3. Tab. 3 shows that all of our proposed affinity measures achieve their best performance when $m_k \approx K$ and the performance degrades when $m_k$ is too small or too large. We conclude that setting $m_k = K$ should be a good choice for spectral clustering.

Table 3: Spectral clustering accuracy with varying cutoff $m_k$: Rand index of spectral clustering results with $K = 10$ clusters, $\mathcal{G} = \mathrm{SO}(2)$ and $k_{\max} = 10$. Each cluster has 50 points. We run 10 trials for all results. For each method we highlight its best result in boldface.

| $p$ | method | Truncation $m_k$ | | | | | |
|---|---|---|---|---|---|---|---|
| | | 2 | 5 | 10 | 20 | 50 | 100 |
| 0.16 | Scalar | $0.828 \pm 0.032$ | $0.847 \pm 0.020$ | $\mathbf{0.865 \pm 0.017}$ | $0.853 \pm 0.014$ | $0.834 \pm 0.010$ | $0.823 \pm 0.012$ |
| | VDM | $0.825 \pm 0.024$ | $0.854 \pm 0.021$ | $0.879 \pm 0.020$ | $0.916 \pm 0.015$ | $\mathbf{0.925 \pm 0.016}$ | $0.912 \pm 0.019$ |
| | Power spec. (ours) | $0.849 \pm 0.022$ | $0.938 \pm 0.018$ | $\mathbf{0.979 \pm 0.008}$ | $0.961 \pm 0.010$ | $0.955 \pm 0.012$ | $0.973 \pm 0.007$ |
| | Opt (ours) | $0.878 \pm 0.025$ | $0.957 \pm 0.016$ | $0.966 \pm 0.010$ | $\mathbf{0.983 \pm 0.007}$ | $0.960 \pm 0.009$ | $0.975 \pm 0.008$ |
| | Bispec. (ours) | $0.869 \pm 0.019$ | $0.948 \pm 0.013$ | $\mathbf{0.975 \pm 0.009}$ | $0.955 \pm 0.014$ | $0.957 \pm 0.008$ | $0.927 \pm 0.016$ |
| 0.20 | Scalar | $0.838 \pm 0.032$ | $0.881 \pm 0.024$ | $\mathbf{0.958 \pm 0.017}$ | $0.941 \pm 0.010$ | $0.845 \pm 0.028$ | $0.830 \pm 0.031$ |
| | VDM | $0.823 \pm 0.027$ | $0.903 \pm 0.015$ | $0.959 \pm 0.011$ | $0.958 \pm 0.011$ | $0.962 \pm 0.008$ | $\mathbf{0.982 \pm 0.005}$ |
| | Power spec. (ours) | $0.894 \pm 0.019$ | $0.985 \pm 0.007$ | $\mathbf{0.997 \pm 0.002}$ | $0.996 \pm 0.002$ | $0.996 \pm 0.002$ | $0.995 \pm 0.003$ |
| | Opt (ours) | $0.905 \pm 0.020$ | $0.993 \pm 0.003$ | $\mathbf{0.998 \pm 0.001}$ | $0.996 \pm 0.001$ | $0.997 \pm 0.001$ | $0.974 \pm 0.008$ |
| | Bispec. (ours) | $0.895 \pm 0.021$ | $0.986 \pm 0.07$ | $\mathbf{0.997 \pm 0.002}$ | $0.996 \pm 0.002$ | $0.964 \pm 0.017$ | $0.917 \pm 0.024$ |
| 0.25 | Scalar | $0.850 \pm 0.016$ | $0.913 \pm 0.018$ | $0.985 \pm 0.008$ | $\mathbf{0.986 \pm 0.009}$ | $0.864 \pm 0.032$ | $0.830 \pm 0.022$ |
| | VDM | $0.854 \pm 0.012$ | $0.950 \pm 0.011$ | $0.992 \pm 0.008$ | $0.993 \pm 0.005$ | $\mathbf{0.993 \pm 0.004}$ | $0.993 \pm 0.005$ |
| | Power spec. (ours) | $0.948 \pm 0.021$ | $0.998 \pm 0.001$ | $\mathbf{1 \pm 0}$ | $\mathbf{1 \pm 0}$ | $\mathbf{1 \pm 0}$ | $\mathbf{1 \pm 0}$ |
| | Opt (ours) | $0.982 \pm 0.008$ | $0.999 \pm 0.001$ | $\mathbf{1 \pm 0}$ | $\mathbf{1 \pm 0}$ | $\mathbf{1 \pm 0}$ | $\mathbf{1 \pm 0}$ |
| | Bispec. (ours) | $0.952 \pm 0.013$ | $0.998 \pm 0.001$ | $\mathbf{1 \pm 0}$ | $\mathbf{1 \pm 0}$ | $\mathbf{1 \pm 0}$ | $\mathbf{1 \pm 0}$ |

**Maximum Frequency $k_{\max}$:** We run another experiment with $K = 10$ clusters, $\mathcal{G} = \mathrm{SO}(2)$, and $m_k = 10$, with $p = 0.16, 0.2, 0.25$. Each cluster contains 50 points. We vary $k_{\max}$ from 2 to 100 and show the Rand indices of clustering results in Tab. 4. We observe that the accuracy gets improved with increasing $k_{\max}$ for all three proposed affinities. However, using a larger $k_{\max}$ increases the computational complexities for all three affinity measures and the dimension of the irrep might increase with $k$ (e.g. the dimension of Wigner $D$-matrix at index $k$ is $2k + 1$), which is undesirable. There is a trade-off between the statistical accuracy and computational complexity. Therefore, we use a moderate $k_{\max} = 10$ in the main paper.

Table 4: Spectral clustering accuracy with varying maximum frequency $k_{\max}$: Rand index of spectral clustering results with $K = 10$ clusters, $\mathcal{G} = \mathrm{SO}(2)$ and $m_k = 10$, each cluster has 50 points. We run 10 trials for all results. For each method we highlight the best result in boldface. Note that the scalar input and VDM are only for $k_{\max} = 0$ and $k_{\max} = 1$, respectively.

| $p$ | method | Maximum frequency $k_{\max}$ | | | | | |
|---|---|---|---|---|---|---|---|
| | | 2 | 5 | 10 | 20 | 50 | 100 |
| 0.16 | Scalar | | | — $0.865 \pm 0$ — | | | |
| | VDM | | | — $0.879 \pm 0$ — | | | |
| | Power spec. (ours) | $0.920 \pm 0.019$ | $0.958 \pm 0.009$ | $0.979 \pm 0.004$ | $0.981 \pm 0.004$ | $0.965 \pm 0.007$ | $\mathbf{0.985 \pm 0.003}$ |
| | Opt (ours) | $0.920 \pm 0.014$ | $0.951 \pm 0.009$ | $0.957 \pm 0.008$ | $0.988 \pm 0.003$ | $0.968 \pm 0.005$ | $\mathbf{0.993 \pm 0.002}$ |
| | Bispec. (ours) | $0.898 \pm 0.025$ | $0.960 \pm 0.010$ | $0.975 \pm 0.008$ | $0.976 \pm 0.007$ | $0.989 \pm 0.005$ | $\mathbf{0.990 \pm 0.004}$ |
| 0.20 | Scalar | | | — $0.958 \pm 0$ — | | | |
| | VDM | | | — $0.959 \pm 0$ — | | | |
| | Power spec. (ours) | $0.991 \pm 0.003$ | $0.974 \pm 0.008$ | $0.997 \pm 0.001$ | $0.997 \pm 0.001$ | $0.999 \pm 0.001$ | $\mathbf{1 \pm 0}$ |
| | Opt (ours) | $0.970 \pm 0.012$ | $0.996 \pm 0.002$ | $0.998 \pm 0.001$ | $0.998 \pm 0.001$ | $\mathbf{0.999 \pm 0.001}$ | $0.999 \pm 0.001$ |
| | Bispec. (ours) | $0.989 \pm 0.005$ | $0.996 \pm 0.002$ | $0.997 \pm 0.001$ | $0.998 \pm 0.001$ | $\mathbf{1 \pm 0}$ | $\mathbf{1 \pm 0}$ |
| 0.25 | Scalar | | | — $0.985 \pm 0$ — | | | |
| | VDM | | | — $0.992 \pm 0$ — | | | |
| | Power spec. (ours) | $0.997 \pm 0.001$ | $\mathbf{1 \pm 0}$ | $\mathbf{1 \pm 0}$ | $\mathbf{1 \pm 0}$ | $\mathbf{1 \pm 0}$ | $\mathbf{1 \pm 0}$ |
| | Opt (ours) | $0.998 \pm 0.001$ | $\mathbf{1 \pm 0}$ | $\mathbf{1 \pm 0}$ | $\mathbf{1 \pm 0}$ | $\mathbf{1 \pm 0}$ | $\mathbf{1 \pm 0}$ |
| | Bispec. (ours) | $0.996 \pm 0.002$ | $\mathbf{1 \pm 0}$ | $\mathbf{1 \pm 0}$ | $\mathbf{1 \pm 0}$ | $\mathbf{1 \pm 0}$ | $\mathbf{1 \pm 0}$ |