[Reviews · NeurIPS 2019]

Reviewer 1



This paper leverages additional assumptions in learning low-dimensional manifolds based motivated by structures seen in cry-EM microscopy imaging and computer vision which involve projections of 3D objects. Specifically, the authors assume a smooth manifold admitting the action of a Lie group G. Using principles from fibre bundles, and defining a invariant moment affinity, they present an optimal alignment algorithm. The paper is well-written, the model is properly motivated and the theory is sound. The design of the algorithm is neat with spectral denoising followed by checking consistency across irreps and moment matching. However, I am not fully convinced with the the improvement in performance especially in cryoEM data. It would be great if the authors could also show examples of classified objects in images and show additional metrics such as F-statistics for evaluating the classification. Comparisons to other manifold learning methods would also strengthen the results.

Reviewer 2



The authors consider unsupervised co-learning on G-manifolds. In particular they extend vector diffusion maps (VDM) to leverage more than one irrep of G. They calculated normalized and (spectrally) filtered adjacency matrices W_k corresponding to each irrep rho_k. They consider three approaches to dealing with pairwise alignment (i.e. finding the group operation that aligns datapoints i and j): 1) explicit optimization 2) using the trace of the power spectrum of W_k, averaged over k 3) using the trace of the bispectra of all pairs of irreps, averaged over pairs. The later two approaches are invariant to the explicit alignment so avoid the inner optimization leading to computational savings. The paper is clearly written although the mathematical content is substantial. There are a number of small typos/grammatical errors so I would suggest carefully proof-reading on resubmission. Is m_k defined in the text somewhere? Define the direct sum when it is first used and not again. It is challenging to fit this much material into an 8 page conference paper and the exposition suffers as a result. For people unfamiliar with topological data analysis more examples/intuition/figures on the setup would be helpful, e.g. a Mobius strip M is a simple canonical example of a fibre bundle where B is a circle and the fibers run perpendicular but twisting to B. Explicitly relating symbols to elements of cryoEM example might also help: e.g. G corresponds to possible rotations of the molecule. This would be more useful than the algorithms which could be moved to the supplement: algos 2, 3 and 4 are trivial (just apply the appropriate equations) and even algo 1 is obvious from the text. This approach has a lot of tunable parameters: k_max, m_k, sigma or number of nearest neighbors. How are these picked? The results are encouraging but it's disappointing that all the applications are to simulated data. Why not apply the method to real cryoEM data? Simply recovering the correct neighbors is not usually a task of interest in its own right: how do these improvements translate into a task of interest? I'm sure what that would be for cryoEM: accuracy of the estimated 3d structure maybe? Typo in fig1: one edge in the second irrep is labelled rho_1 rather than rho_2. Edit: I've read the author response and the other reviews. I'm updating my score to a 7.

Reviewer 3



The paper builds on the work in [21], which dealt only with SO(2). This paper adds multiple representations as a "multiview learning" problem and the bispectrum, which are nice additions and sufficient novelty. The mathematics are technically sound and elegantly based on group representation theory. I believe more writing should be spent in explaining more background on representation theory and motivating the intuition behind it. Section 3, giving background on principal bundles, is very nicely written. The next section dives into irreducible representations, which I think the average NeurIPS reader will not be familiar with. A few sentences defining and motivating representation theory would improve the clarity. I realize that a full tutorial on representation theory is not practical. I have some familiarity with representation theory, but I was unfamiliar with Wigner bases and Clebsch-Gordan coefficients. Some more explanation (again, at an intuitive level) there would be helpful. In a related point, I am having trouble understanding what you gain by incorporating multiple representations over using a single "best" representation. I can see that the proposed work will be significant in cryo-EM and possibly in computer vision (accounting for transformations of objects being detected in images), but the applicability there is less clear.

[Author Response · NeurIPS 2019]

We thank the reviewers for their careful reading and valuable comments, which we address one by one below.

**Reviewer 1: Q1:** *Show comparisons to other manifold learn-*
*ing methods, additional metrics, and classified objects.* **R:** We
add the comparisons with Diffusion maps (DM) (see Fig. 1)
and Laplacian eigenmaps (LE). The performance of LE is sim-
ilar to DM, since the data are uniformly distributed on the
manifold. In the first two experiments in Sec. 5 of our paper,
we focus on the accuracy of the nearest neighbor identification
given extremely noisy initial graph structure. Fig. 1 shows the
geodesic distances of the estimated nearest neighbors on $S^2$.

Figure 1: Histograms of $\arccos \langle v_i, v_j \rangle$ between identified nearest neigh-
bors. *Left*: simulated data for $\mathcal{M} = SO(3)$ under random rewiring model;
*Right*: cryo-EM images. The clean histogram should peak at small angles.

We use the **Jaccard index** as an additional metric. The Jaccard
index evaluates the similarity of the estimated and the true nearest neighbors of each node. For the synthetic data on $S^2$
at $p = 0.1$, the mean Jaccard indices are *0.196 (Power Spec.), 0.209 (Bispec.), **0.215 (Opt.)**, 0.059 (VDM), 0.042 (DM)*
(higher the better). For the cryo-EM images (SNR= 0.005), they are *0.033(Power Spec.), **0.035 (Bispec.)**, 0.031 (Opt.),*
*0.028 (VDM), 0.024 (DM)*. We will add these results in revision. The histogram contains the information on how close
the estimated nearest neighbors are, whereas the Jaccard index only measures the set similarity. For the application in
spectral clustering, we use **rand index** to measure the performance in the paper. The $F$-score does not apply here since
the examples are not binary (or multi-class) classification problems with labels. We will clarify the choice of metrics for
the performance evaluation in revision. We will add more illustrations to show samples of estimated nearest neighbor
images and improvement in image denoising (see response to R2 Q3 and Fig. 2).

**Improvements:** *More comprehensive evaluation of performance especially on real data.* **R:** We will add additional
metric, comparisons, and illustrations mentioned in the response to R1 Q1 in the revised manuscript.

**Reviewer 2: Q1:** *Small typos and grammatical errors, $m_k$, direct sum.* **R:** Thanks for pointing these out, we will
correct the typos/errors and clarify the definition of parameters in our revised manuscript.

**Q2:** *Tunable parameters?* **R:** The choice of parameters was explained in the captions of Figs. 2 and 3 of the paper; we
will discuss them in greater detail in the revised version of the main paper. The maximum frequency $k_{\max}$ is chosen to
be as large as possible within our computational budget—this is because it is empirically observed that the performance
improves as $k_{\max}$ gets larger, but saturates once $k_{\max}$ becomes sufficiently large. The parameter for the number of
eigenvectors $m_k$ is chosen relatively small ($\leq 50$) for computational efficiency and to exclude the noise-sensitive
"high-frequency" eigenvectors. For nearest neighbor searching, the number of nearest neighbors is chosen to ensure a
well connected sparse graph in the noise-free setting. For spectral clustering, the initial graph is given and fixed.

**Q3:** *Application to the real cryo-EM data?* **R:** There
is no direct way to compare the performance of nearest
neighbor identification algorithms on real microscope
images, since their viewing angles and underlying clean
images are unknown. We used simulated data in our
experiments so that the outputs can be compared and
contrasted with the "ground truth." Nevertheless, as
a proxy to real data experiments, we will add results
demonstrating how the denoising step can benefit from

(a) Clean      (b) Noisy      (c) Init.      (d) VDM      (e) Bispec.

Figure 2: (a) Clean projections of 70S ribosome; (b) Noisy images with SNR $= 0.01$;
(c) to (e) Denoised images based on the graph and alignments identified by the initial es-
timation, VDM and Bispectrum-like affinity (this paper). MSEs of the denoised images
are (c) 6.24, (d) 5.72, and (e) **4.97** (lower is better).

the improved nearest neighbor identification (see Fig. 2); it is known that the quality of the denoised images directly
contributes to the 3D reconstruction results (see e.g. explanation in reference [74] of the main paper).

**Improvements:** *Exposition and application.* **R:** We will move the algorithms into the supplement and add more
explanations and intuition in the main paper. Our paper provides a framework for analyzing data that lie on or close to a
manifold with a group action and is not limited to cryo-EM problem, e.g., spectral clustering with $SO(2)$ and $SO(3)$
transformations. We will add the cryo-EM denoising results in revision. Other tasks will be explored in the future.

**Reviewer 3: Q1:** *More background and intuition.* **R:** We will move Alg.2–4 to the supplementary material, and add
more explanation on group theory and irreducible representations in the main paper. We will also provide motivating
examples with $SO(3)$ to explain the intuition of using Wigner D-matrices and Clebsch–Gordan coefficients.

**Q2:** *Gain of incorporating multiple representations over the "best" representation?* **R:** In practice, observations from
real data—in any representation—always contain certain level of noise, even for the "best" representation. Incorporating
multiple representations allows us to leverage the inherent consistency across different representations of the same
information to better remove noise. Methodologically, incorporating multiple representations creates a "redundant"
representation akin to redundant wavelets/frames/dictionaries in applied harmonic analysis, which are known to be
more robust to noise due to the additional structural rigidity. We will further clarify this in the revised version.

**Q3:** *Clarify the applicability in cryo-EM and computer vision.* **R:** We will clarify in writing how the proposed work
can be applied in cryo-EM and computer vision, and add more illustrations such as Fig. 2.

**Improvements:** *Remove Algs. 2–4. Write more about the background and intuition.* **R:** Please see response to R3 Q1.

[Meta-Review · NeurIPS 2019]

This paper proposes a novel multi-view framework for alignment affinity in a G-manifold with extending to multiple irreps of G. The reviewers agree that the paper has interesting algorithmic contributions and applications to Cryo-EM. The author’s rebuttal has been partially convincing. Further results on NN denoising as mentioned in the rebuttal, in the final revision would be crucial.